# Policy Iteration with Gaussian Process based Value Function Approximation

Ashwin Khadke
The Robotics Institute
Carnegie Mellon University
Pittsburgh, PA 15213
Email: akhadke@andrew.cmu.edu

Akshara Rai
Facebook AI Research
Menlo Park, CA 94025
Email: akshararai@fb.com

*Abstract*—In this work, we explore the use of Gaussian processes (GP) as function approximators for Reinforcement Learning (RL), and build estimates of the value function and Q-function using GPs. Such a representation allows us to learn Q-functions, and thereby policies, conditioned on uncertainty in the system dynamics, and can be useful in sample efficiently transferring policies learned in simulation to hardware. We use two approaches GPTD and GPSARSA, from Engel et al. [1] to build approximate value functions and Q-functions, respectively. While for simple, continuous problems, we found these to be effective at approximating the value function and the Q-function, for discontinuous landscapes GPSARSA deteriorates in performance, even on simple problems. As the problem complexity increases, for example, for an inverted pendulum, we find that both approaches are extremely sensitive to the GP hyperparameters, and do not scale well. We experiment with a sparse variant of the algorithm, but find that GPSARSA still converges to poor solutions. Our experiments show that while GPTD and GPSARSA are nice theoretical formulations, they are not suitable for complex domains without extensive hyperparameter tuning.

## I. Introduction

Reinforcement learning has shown great promise and success in discrete problems like such as AlphaGo by Silver et al. [9], as well as high-dimensional continuous problems, like Lillicrap et al. [5]. In general, for continuous state and action spaces, RL algorithms use function approximators for approximating the value function, as well as the policy. Common function approximators used in RL include neural networks [3], or linear basis functions [11], or Gaussian Processes [2]. However, the sample-efficiency of RL algorithms for continuous problems can be very poor, partially due to the data-hungry nature of function approximators, like neural networks, making their deployment on real-world problems hard.

Especially for robotics problems, near-exhaustive search in the large state and action space of robots is infeasible. In such cases, using simulation for warm-starting the approximation of the value function, Q-function or policy can be beneficial. However, due to mismatch between simulation and hardware, policies learned in simulation do not successfully transfer to hardware, especially for complex dynamical systems like legged robots, as pointed out by Li et al. [4]. An alternative in literature is to learn conservative policies using domain

randomization as in Tobin et al. [13], by randomizing the physics and noise parameters of the simulator. Such policies can be successful on hardware directly, as shown by Tan et al. [12] but also might fail, as Yu et al. [14] point out. In cases of failure, there isn't a 'recovery' strategy that can lead to reasonable learning of policies on hardware.

Instead of learning policies robust to dynamics uncertainties, we explore the possibility of learning a Q-function in simulation, and sample efficiently adapting it on hardware. This can be used to generate a policy by maximizing the Q-function at each state. Such an approach can recover from cases where the policy learned in simulation fails on hardware.

For effectively updating the Q-function online, we identified two key characteristics in our chosen function approximator:

- An estimate of the accuracy of the Q-function in different parts of the space : Assuming that the simulation is inaccurate around certain states, example involving contacts, but accurate around others, like during free-space motion, an estimate of the accuracy of the simulated Q-function could be used for guiding the optimization on hardware.
- Quick update and reliable extrapolation on hardware : Once new data is gathered on hardware, we would like that the simulated Q-function can be updated quickly and sample-efficiently, and extrapolates to other states that have not been seen on hardware yet. Moreover, the policy should be updated fast based on updates to the Q-function.

With both these considerations, we decided to use a Gaussian Process (GP) for approximating our Q-function. To imprve sample-efficiency, a Q-function learned in simulation can serve as prior for learning on hardware. To investigate the feasibility of using GPs as function approximators we experimented with the GPTD and GPSARSA algorithms from Engel et al. [1]. GPTD is a temporal-difference (TD) learning based algorithm for computing the value function of a given policy. GPSARSA builds a GP based estimate of the optimal Q-function, in a fashion similar to SARSA from Rummery and Niranjan [8]. We describe these methods in Section II.

In our initial experiments we used grid world environments with discrete state and action space (Fig. 1). We observed that both GPTD and GPSARSA could approximate the optimal value function when the value function is smooth, as shown

in Figure 3c. However, for more complex discrete problems, like grid world with holes, we observed that GPSARSA had trouble approximating the optimal Q-function. As we moved to continuous domains, we observed that both GPTD and GPSARSA could be highly sensitive to hyperparameter selection, and performed significantly worse than exact value function computation. When we learned GP hyperparameters from the true value-function, and used these for learning, we observed that we could converge to the true value. But using different, but close, hyperparameters would lead to much poorer learning. In retrospective, the GPTD and GPSARSA approaches have only been tested on simplistic 2D navigation environments, and did not generalize to more complex dynamical systems. While this is a nice theoretical formulation, the applicability to practical problems is limited.

## II. BACKGROUND

### A. Temporal difference learning with Gaussian processes (GPTD)

GPTD uses Gaussian Processes (GPs) to construct an estimate of the value function for a given fixed policy $\pi$ using TD learning. The algorithm uses the Bellman equation (Eq. (1)) to build a posterior of the value function $V$

$$r(\boldsymbol{s}) = V^\pi(\boldsymbol{s}) - \gamma V^\pi(\boldsymbol{s}') \qquad (1)$$

from observed rewards $r$ for current state $\boldsymbol{s}$ and next state $\boldsymbol{s}'$. Assuming a Gaussian prior $\mathcal{N}(0, \Sigma)$ over the value function, the posterior after $t$ time-steps, on observing a sequence of rewards $\boldsymbol{r}_{t-1} = [r_1, r_2, \cdots r_{t-1}]$ can be described as in Eq. (2). Refer to Engel et al. [1] for details.

where,
$$k(\boldsymbol{s}, \boldsymbol{s}') = \sigma_f^2 e^{(\boldsymbol{s}-\boldsymbol{s}')\Sigma_{\text{SqExp}}^{-1}(\boldsymbol{s}-\boldsymbol{s}')^T}$$
$$k_t(\boldsymbol{s}) = \begin{bmatrix} k(\boldsymbol{s}, \boldsymbol{s}_1) & \cdots & k(\boldsymbol{s}, \boldsymbol{s}_t) \end{bmatrix}^T$$
$$K_t = \begin{bmatrix} k(\boldsymbol{s}_1, \boldsymbol{s}_1) & \cdots & k(\boldsymbol{s}_1, \boldsymbol{s}_t) \\ \vdots & & \vdots \\ k(\boldsymbol{s}_t, \boldsymbol{s}_1) & \cdots & k(\boldsymbol{s}_t, \boldsymbol{s}_t) \end{bmatrix} \qquad (2)$$
$$H_t = \begin{bmatrix} 1 & -\gamma & 0 & \cdots & 0 \\ 0 & 1 & -\gamma & \cdots & 0 \\ \vdots & & & & \vdots \\ 0 & 0 & \cdots & 1 & -\gamma \end{bmatrix}$$

The kernel function $k$ characterizes an interpolation scheme which together with the length scale $\Sigma_{\text{SqExp}}$ determines how different points in the GP influence the prediction $V$ at a new query state $\boldsymbol{s}$.

However, the posterior update can quickly become intractable if every encountered state $\boldsymbol{s}_t$ is added to the GP. To mitigate this, GPTD maintains a dictionary of 'unique' states $\mathcal{D} = \{\tilde{\boldsymbol{s}}_i | i \in \{1, \cdots, n\}\}$ which are used as a set of basis vectors for the entire state space. $\mathcal{D}$ is initialized to be empty and a state $\boldsymbol{s}$ is added to $\mathcal{D}$ if the projection error $(\min_a k(\boldsymbol{s}, \sum_i a_i \tilde{\boldsymbol{s}}_i))$ is greater than a threshold $\nu$. Let $\mathcal{D}_t$ be

the dictionary at time-step $t$ and $A_t$ of size $(t \times |\mathcal{D}_t|)$ the projection matrix. The $j^{\text{th}}$ row in $A_t$ denotes the projection coefficients of state $\boldsymbol{s}_j$ onto the states in $\mathcal{D}_t$. The posterior update of the value function given the dictionary $\mathcal{D}_t$ is described in Eq. (3).

$$V^\pi(\boldsymbol{s}) | \boldsymbol{r}_{t-1} \sim \mathcal{N}\Big(\tilde{k}_t(\boldsymbol{s})^T \tilde{H}_t^T (\tilde{H}_t \tilde{K}_t \tilde{H}_t^T + \Sigma)^{-1} \boldsymbol{r}_{t-1},$$
$$k(\boldsymbol{s}, \boldsymbol{s}) - \tilde{k}_t(\boldsymbol{s})^T \tilde{H}_t^T (\tilde{H}_t \tilde{K}_t \tilde{H}_t^T + \Sigma)^{-1} \tilde{H}_t \tilde{k}_t(\boldsymbol{s})\Big)$$

where,
$$\tilde{k}_t(\boldsymbol{s}) = \begin{bmatrix} k(\boldsymbol{s}, \tilde{\boldsymbol{s}}_1) & \cdots & k(\boldsymbol{s}, \tilde{\boldsymbol{s}}_{|\mathcal{D}_t|}) \end{bmatrix}^T$$
$$\tilde{K}_t = A_t^T K_t A_t$$
$$\tilde{H}_t = H_t A_t$$
$$(3)$$

We compute this posterior after each episode. Note that, GPTD computes the value function $V^\pi_{\text{GPTD}}$ for a given policy $\pi$ and does not actually compute the optimal policy $\pi^*$. In our experiments we compute $V^{\pi^*}_{\text{GPTD}}$ using the optimal policy obtained from policy iteration. We compare the mean of the GP approximating $V^{\pi^*}_{\text{GPTD}}$ with $V^*$ to estimate its accuracy. In future text, we drop $\pi^*$ for simplicity of notation and refer to $V^{\pi^*}_{\text{GPTD}}$ as $V_{\text{GPTD}}$

### B. Approximating Q-function with GPs

In similar spirit to TD learning, the SARSA algorithm [8] constructs Q-function estimates from the observed rewards (Eq. (4)).

$$Q(\boldsymbol{s}, \boldsymbol{a}) = r(\boldsymbol{s}, \boldsymbol{a}) + \gamma Q(\boldsymbol{s}', \boldsymbol{a}') \qquad (4)$$

GPSARSA builds a GP based estimate of the Q-function using this relation. The approach is similar to GPTD except GPSARSA constructs a GP over states and actions, and uses the current Q-function estimate to sample actions during roll-outs (Eq. 5). We sample

$$\pi(\boldsymbol{s}) = \underset{\boldsymbol{a}'' \sim [\boldsymbol{a}_{\min}, \boldsymbol{a}_{\max}]}{\arg\min} Q(\boldsymbol{s}, \boldsymbol{a}'')$$
$$\text{where } Q(\boldsymbol{s}, \boldsymbol{a}'') \sim \mathcal{N}(\mu(\boldsymbol{s}, \boldsymbol{a}), \Sigma(\boldsymbol{s}, \boldsymbol{a})) \qquad (5)$$

a fixed number of actions from a continuous range to evaluate the minima in Eq. (5). Furthermore, we compute the value function for the optimal policy as shown in Eq. (6).

$$V_{\text{GPSARSA}}(\boldsymbol{s}) = \underset{\boldsymbol{a}'' \sim [\boldsymbol{a}_{\min}, \boldsymbol{a}_{\max}]}{\min} \mu(\boldsymbol{s}, \boldsymbol{a}'') \qquad (6)$$

## III. EXPERIMENTS

### A. Experimental setting

The environments studied in our experiments were:

1) **Grid world**: An environment with discrete state and action space shown in Fig. 1. The objective is to reach the goal without landing into a hole. The agent is initialized at a random location, and in each step can choose to move up, down, left or right. It receives a reward of +1 on reaching the goal and -1 on landing in a hole.

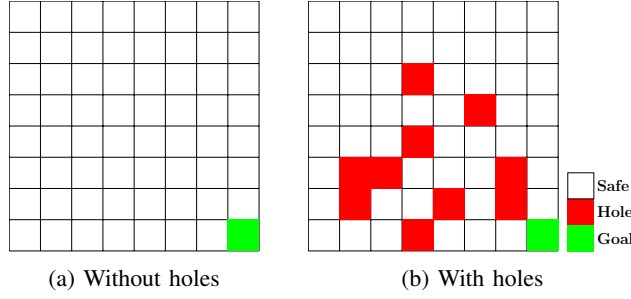

(a) Without holes      (b) With holes

Safe
Hole
Goal

Fig. 1: Grid world environment.

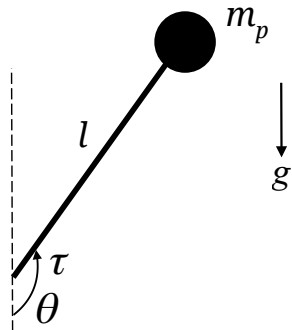

Fig. 2: Pendulum with mass $m_p = 1$kg, length $l = 0.9$m under the influence of gravity ($g = 9.81$ms$^{-2}$) and input $\tau$ such that $|\tau| \leq 9$Nm

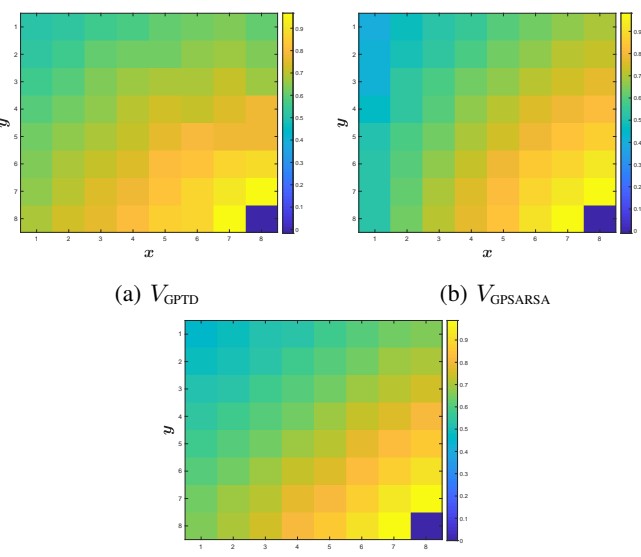

(a) $V_{\text{GPTD}}$      (b) $V_{\text{GPSARSA}}$

(c) Optimal value function ($V^*$) obtained using vanilla policy iteration

Fig. 3: Value functions for grid world environment without holes depicted in Fig. 1a

| Environment | $\sum_{s \in \mathcal{S}} |V - V^*|/|\mathcal{S}|$ | |
|---|---|---|
| Algorithm | Without holes | With holes |
| GPTD | 0.0016 | 0.046 |
| GPSARSA | 0.064 | 0.34 |

TABLE I: Mean error in value functions for the grid world environments computed using GPTD and GPSARSA

2) **Inverted pendulum**: The system is shown in Fig. 2, and the objective is to design a control policy for $\tau$ to swing up the pendulum to an upright position ($\theta = \pi$) under gravity. This is an environment with a continuous state ($\boldsymbol{s} = [\theta, \dot{\theta}]$) and action ($\boldsymbol{a} = \tau$) space. We use a quadratic cost function $(\boldsymbol{s}-[\pi,0])Q(\boldsymbol{s}-[\pi,0])^T + \boldsymbol{a}R\boldsymbol{a}^T$ where $Q = \text{diag}([25, 0.02])$ and $R = 0.001$, to train policies.

We compared the efficacy of the following approaches at approximating the value function of the environments described above:

1) **Policy Iteration**: We use a tabular representation for the optimal policy ($\pi^*$) and value function ($V^*$). For the inverted pendulum environment, we discretize the state space into a grid and sample actions from a continuous range to compute a policy.

2) **GPTD with optimal policy**: Using $\pi^*$ learned from policy iteration, we compute a GP based representation ($V_{\text{GPTD}}$) for $V^*$.

3) **GPSARSA** : We compute a GP based representation for the optimal Q-function and compare the resulting value function ($V_{\text{GPSARSA}}$) with $V^*$ by maximizing the Q-function at discretized states.

### B. Experimental results

*1) Frozen lake:* Fig. 3c depicts $V^*$ for the grid world environment without holes shown in Fig. 1a. ($V_{\text{GPTD}}$) and ($V_{\text{GPSARSA}}$) are show in Fig. 3a and Fig. 3b respectively. In this experimental setting, both GPTD and GPSARSA are effective at learning the optimal policy and value function. Without holes the value function and Q-function are continuous and smooth, and the GPs are good at estimating them (Fig. 3).

Next, we obtained value function estimates for the environment with holes, shown in Fig. 1b. The qualitative results are shown in Fig. 4. Table I compares the accuracy of the two algorithms in estimating the optimal value functions. Despite the sharp discontinuities in $V^*$, GPTD computes a reasonable estimate for the optimal value function (Fig. 4a). However, GPSARSA fails to converge to the optimal Q-function (Fig. 4b).

*2) Inverted pendulum:* The optimal value function for inverted pendulum is shown in Fig. 5e. We use the squared exponential kernel (Eq. (3)) with different length-scales per dimension for the GP ($\Sigma_{\text{SqExp}} = \text{diag}([\sigma_\theta, \sigma_{\dot{\theta}}])$ for GPTD and $\Sigma_{\text{SqExp}} = \text{diag}([\sigma_\theta, \sigma_{\dot{\theta}}, \sigma_\tau])$ for GPSARSA). The length scales $\sigma_\theta$, $\sigma_{\dot{\theta}}$ (and $\sigma_\tau$) describe how close a state (or state-action pair) has to be to a query point along the corresponding dimensions to influence the interpolated value. We observe the quality of fit to be highly sensitive to the choice of kernel parameters ($\Sigma_{\text{SqExp}}, \sigma_f$).

With manually tuned kernel parameters based on the state discretization resolution used in policy iteration, we observe

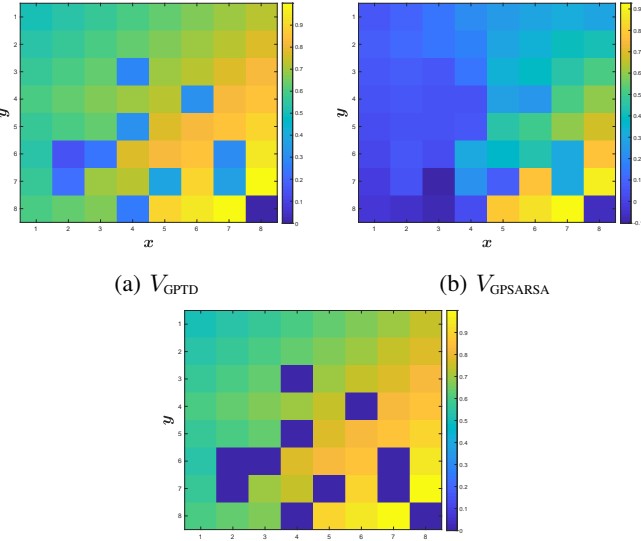

(a) $V_{\text{GPTD}}$        (b) $V_{\text{GPSARSA}}$

(c) Optimal value function ($V^*$) obtained using vanilla policy iteration

Fig. 4: Value functions for frozen lake environment with holes depicted in Fig. 1b

poor performance with GPTD. Fig. 5a depicts the value function estimate obtained using GPTD in this case. Next, we optimized the kernel parameters by maximizing the marginal likelihood of observing $V^*$ at a few sampled points in the state-space. With tuned kernel parameters GPTD approximates the optimal value function quite well (Fig. 5b).

We also attempted to learn the optimal Q-function using GPSARSA with little success (Fig. 5d and 5c), both with optimized and hand-tuned parameters. These experiments demonstrated that the GPSARSA algorithm has even poorer convergence properties than GPTD, even with carefully tuned kernel parameters.

*3) Sparse GPs with inverted pendulum:* Our experiments with GPTD and GPSARSA showed that the kernel parameters not only affect the quality of fit but also the amount of samples required to build these estimates. The GPTD and GPSARSA algorithms construct a dictionary of states to fit the GP using the kernel-based distance metric to decide which state samples to incorporate in this dictionary (Section II). Therefore, a poor choice of kernel parameters could lead to a really large sample set which slows down learning, or a very small sample set which degrades the quality of fit. To overcome these problems, we experimented with Sparse Gaussian Processes using Pseudo Inputs (SPGPs) [10]. SPGPs compress the number of data points in a GP with a set of representative pseudo points. The location and values of these pseudo points along with the kernel parameters are computed such that they maximize the marginal likelihood of the observed data. In literature, Martin et al. use SPGPs for computing value functions for parameterized policies.

We attempted to use SPGPs in the GPTD and GPSARSA

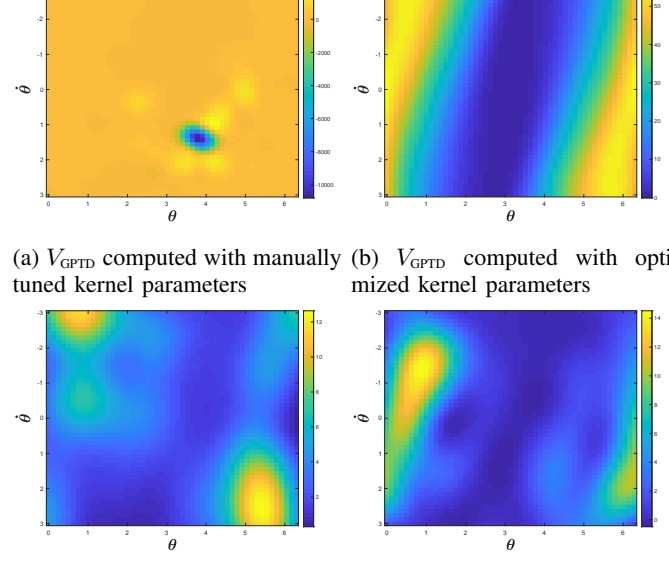

(a) $V_{\text{GPTD}}$ computed with manually tuned kernel parameters    (b) $V_{\text{GPTD}}$ computed with optimized kernel parameters

(c) $V_{\text{GPSARSA}}$ computed with manually tuned kernel parameters    (d) $V_{\text{GPSARSA}}$ computed with optimized kernel parameters

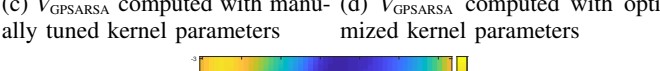
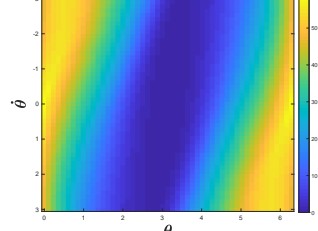

(e) Optimal value function ($V^*$) obtained using vanilla policy iteration

Fig. 5: Value functions for the pendulum system

| Algorithm | $\sum_{s \in \mathcal{S}} |V - V^*|/|\mathcal{S}|$ | | |
| | Optimized parameters | Hand tuned parameters | SPGP |
|---|---|---|---|
| GPTD | 0.41 | 168 | 1.68 |
| GPSARSA | 20.8 | 20.4 | 6 |

TABLE II: Mean error in value functions for the pendulum system computed using GPTD and GPSARSA

algorithms. If the GP exceeds a certain size, we sparsify it and continue with vanilla GPTD/GPSARSA treating the sparsified GP as a prior over the value function/Q-function. Fig. 6 showcases the resultant value functions. Table II summarizes the quantitative results. Our experiments and the ensuing results highlight the sensitivity of GPTD and GPSARSA to kernel parameters. Finding suitable parameters without knowing the value function or Q-function landscape is very difficult and an iterative procedure to refine them, such as SPGP, seems to help. GPSARSA uses the Q-function to infer the policy which is computationally expensive especially as the size of the GP grows. Instead, using an explicit representation for the policy would provide a speedup in learning.

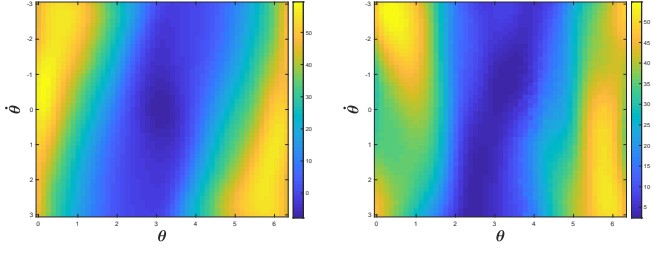

| (a) Obtained using GPTD | (b) Obtained using GPSARSA |

Fig. 6: Sparse Gaussian Processes using Pseudo Inputs (SPGP) based value function estimates for the pendulum system .

## IV. CONCLUSION

In this work, we experimented with Gaussian Processes (GPs) as function approximators in RL. To build a value function and a Q-function using GPs, we used the GPTD and GPSARSA algorithms from [1]. In problems where we expect the optimal value function to be continuous and smooth, both algorithms perform well. However, when discontinuities arise, as in the case of the grid world environment with holes, GPSARSA converges to poor solutions. For an inverted pendulum, a continuous space problem, both GPTD and GPSARSA prove to be highly sensitive to kernel parameters.

Through our experiments we could identify two main bottlenecks to using GPs as function approximators. First, the sensitivity to chosen kernel function. Second, inference is computationally expensive as size of the GP grows. The first bottleneck can be addressed by choosing domain informed kernel functions [7] or by tuning kernel parameters. However, for a general problem appropriate kernel functions are difficult to find. Furthermore, tuning kernel parameters is difficult without knowing the value function landscape apriori. The second bottleneck makes it difficult to use GPs for implicitly representing policies, such as in GPSARSA. To tackle this issue, building an explicit representation for the policy while learning a GP based estimate for the value function can be useful.

Using GPs to represent value functions provides uncertainty estimates that may be useful in improving exploration. However, the above two bottlenecks limit its applicability to complex continuous control problems.

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
