# OpenReview forum: "Policy Iteration with Gaussian Process based Value Function Approximation"
_roboticsfoundation.org/RSS/2020/Workshop/RobRetro — RobRetro 2020_

### Official Review · AnonReviewer1 · 2020-06-24
**Retrospective on using GPs for Value Function Approximation**

**Confidence:** 5
**Rating:** 8

**Review:**

This paper takes a look and presents (positive+negative) results on using GPs for Value Function approximation. The insights given on when the two considered approaches work, vs when they do not work will be of interest of researchers trying to work on similar problems, making this a good retrospectives paper.

Some considerations for the authors (and maybe for the final version):
1) I'd be curious to know what you think the next step should be? Is your conclusion then that GPs are just not realistic then for real world problems?
2) Would other GP kernels help with discontinues problems?

---

### Decision · Program_Chairs · 2020-06-25

Accept